# Changes in Fruit Quality Phytochemicals of Late-Mature Peach ‘Yonglian No.1’ during Storage

**DOI:** 10.3390/molecules27196319

**Published:** 2022-09-25

**Authors:** Wen Li, Zekang Pei, Juane Shang, Hongjuan Yang, Xiaohua Kui, Zhifang Zeng, Cuilan Ma, Dongliang Qiu

**Affiliations:** 1College of Horticulture, Fujian Agriculture and Forestry University, Fuzhou 350002, China; 2Economic Crop Technology Promotion Station, Gutian Agricultural Bureau, Ningde 352200, China

**Keywords:** phenolic, anthocyanin, flavonoid, TSS, late-mature peach fruit

## Abstract

In this study, the changes in quality parameters and sensory-influencing parameters from the peel, red flesh, and white flesh of ‘Yonglian No.1’ peach fruits were analyzed during cold storage. The results indicated that the contents of total soluble solids (TSS), soluble sugar, organic acid, vitamin C, total anthocyanin, phenol, and flavonoids, as well as the good fruit rate varied depending on the storage stages and storage treatments. The peach fruits in MAP stored for 50 days had favorable exterior qualities, a good fruit rate of 100%, and a higher content of total soluble solids (TSS) at 12.6%. MAP was significantly effective at maintaining fruit firmness, the content of TSS, soluble sugar, organic acid, vitamin C, total anthocyanin, phenol, and flavonoids. Among the derivatives of anthocyanin, both cyanidin and pelargonidin were found in the peel, with a content of 33.45 mg/kg FW and 1.82 mg/kg FW, respectively. However, cyanidin was detected in the flesh with a content of 40.42 mg/kg FW. In the present work, the differences regarding phytochemical profiles and physical properties were mainly correlated with the storage stages and storage treatments of peach fruit. ‘Yonglian No.1’ had higher levels of health-promoting compounds during storage and maintained favorable quality.

## 1. Introduction

As an excellent functional food, fruits are high in antioxidants and nutritional compounds, and their naturally occurring substances also have health-promoting properties [1].

Peach (*Prunus persica* L.) is a rich source of many phytochemicals, including vitamins, phenolic compounds, organic acid, etc. [2]. Anthocyanin is a water-soluble pigment and natural colorant, being one of the reasons that fruits, flowers, and vegetables show colors [3]. In addition, it is known that these compounds play roles as health promoters which have features such as antioxidant, antimicrobic, antidiabetic, anti-inflammatory, antihypertensive, anticancer and immunoprotective effects. Also, these characteristics are known to have indirect impacts, for instance, on the cardiovascular system and in preventing neurodegeneration [4]. Previous reports have indicated that the major anthocyanin in peach fruit is cyanidin-3-glucoside [5].

At present, most peach varieties in Fujian, China, are small mid-late and late-mature cultivars which mature between June and July. As the last batch of a mature fresh peach in the Fujian market, ‘Yonglian No.1’ peach fruit start to ripen in the last part of August, in Fujian province. It is a fresh fruit in the off-season, rich in vitamin C and phenolic compounds [6]. Peaches are climacteric fruit which rot at ambient temperatures, and the shelf life of peach fruit is generally only 5 to 7 days [7]. After harvest, fruit quality is highly curtailed due to flesh browning, rotting, mollification, all of which negatively affect their value [8,9,10]. Therefore, maintaining the levels of the health-promoting compounds during postharvest storage has important implications for human health.

Packaging and storage conditions are important factors which affect the physiological changes in peaches after harvest. Both are of great significance to the storage and sale of peach fruit. The average storage of ‘Qinchao’ peaches at 4 °C using a polyethylene (PE) bag was 58 days [11]. Modified atmosphere packaging (MAP) is used to replenish low-temperature regulation by lowering O_2_ levels and increasing CO_2_ and humidity levels around the fruit or other produce. This has the effect of delaying ripening, suppresses the decay of fresh fruit, inhibits the degradation of chlorophyll and other pigments, and maintains the organoleptic properties and fresh-like characteristics of the fruit [12,13,14]. This could also have an influence on reducing the loss of weight and could prevent the postharvest decay of ‘Douradão’ peaches [15].

Studies on the effects of storage treatments on the health-promoting compounds in ‘Yonglian No.1’ peach fruits, as well as on the maintenance of standard and sensory-influencing qualities of peach fruits will provide key information for maintaining phytochemical contents. This study investigated the influence of 4 ± 1 °C cold storage, modified atmosphere packaging (MAP) and air-permeable packaging (AP) conditioning on the levels of health-promoting compounds in ‘Yonglian No.1’ peach fruits, examining the levels of good fruit rate, fruit firmness, flesh browning rate, total soluble solids (TSS), soluble sugar, organic acid, vitamin C, anthocyanin, total phenol, and flavonoid activity, during a 50-day storage period.

## 2. Materials and Methods

### 2.1. Fruit Materials, Standard Samples, Package, and Storage Condition

The ‘Yonglian No.1’ peach fruits were harvested from the Gantang orchard in Ningde City, Fujian Province on 19 August 2019. In this study, approximately 600 peach fruits, uniform in size and firmness, 80% majority (the green color of fruit apex began to fade obviously, while the red color on the fruit surface began to appear and its flesh retained some firmness) [16], with no diseases, insect pests or mechanical injury were selected as the storage material; fruits from each of three peach trees in four directions of east, west, south, and north were harvested from the same orchard and transported to the laboratory on the same day. The age of the peach trees was 5 years, and the plants were spaced regularly, about 5 m × 5 m apart.

The peach fruits (day 0) were randomly divided into two factors (Fruit washing by tap water or not, and PE bag with pores or not), four treatments in total, and treated as soon as possible after the fruits were transported to the laboratory. One group was immersed in tap water for 30 min, then laid out and dried for 24 h at room temperature (26 °C). For the AP, nine pores were punched uniformly on the surface of the PE bag (280 mm × 400 mm, 9.40 g, 0.03 mm thick, Heyuan Evergreen Plastic MFG. CO., LTD., Heyuan, China) with a punch (8 mm diameter of each); For the MAP, no pores were punched on the surface of the PE bag. Packaging batches for each treatment were as follows:

A treatment (MAP): peach fruit without washing and the PE bag without pores;

B treatment (AP): peach fruit without washing and the PE bag with pores;

C treatment (MAP + Washing): peach fruit washed with tap water and the PE bag without pores;

D treatment (AP + Washing): peach fruit washed with tap water and the PE bag with pores.

‘Yonglian No.1’ peach fruits were stored at 4 ± 1 °C. At every different time point (0, 10, 20, 30, 40, and 50 days) for each group, 27 fruit samples in three biological replicates of nine fruit each were taken for the analysis of fruit quality. After the above determination, the peel from the equator of the peach fruit (Figure 1a), red flesh (Figure 1b), and white flesh (Figure 1c) from the same sample were collected, immediately frozen with liquid nitrogen and stored at −80 °C for further analysis.

### 2.2. Determination of Storage Good Fruit Rate, Fruit Firmness, and Flesh Browning Rate

The good fruit rate (Figure 2a) was calculated: good fruit rate (%) = (number of flawless fruits/total number of fruits) × 100%. Fruit without disease or mechanical injury were regarded as flawless fruit.

Fruit firmness was measured using a fruit firmness tester (GY-3, Honeywell Inc., Charlotte, NC, USA) on each fruit equator after the removal of a 1 mm-thick slice of peel. The results were expressed in kg/cm^2^. Three biological replicates were used for each sample.

Before deciding on the fruit quality parameters, 27 fruits as one replicate were used to study the above indicators.

The flesh browning rate (Figure 2b) was calculated: flesh browning rate (%) = (number of browned fruits/total number of fruits) × 100%.

### 2.3. Determination of Total Soluble Solids (TSS), Soluble Sugar, Organic Acid, and Vitamin C

Nine peaches were randomly picked and squeezed by hand pressing. The pressed juices were dropped directly onto a hand-held refractometer (model N1t, Atago Co., Tokyo, Japan) to measure the TSS, and the results were expressed as a %.

The content of soluble sugar was tested using Anthrone Colorimetry, following the method previously detailed in [17]. The absorbance was recorded at 630 nm wavelength using a UV5100H UV-V spectrophotometer (Yuan Analytical Instrument Co., Ltd., Shanghai, China) and the results were expressed as a %. The results were expressed as the mean of three replicates.

The content of organic acid was tested using sodium hydroxide titration, following the method previously detailed in [18]. Three biological replicates were used for each sample, and the organic acid content was expressed as a %.

Vitamin C was determined by using the 2, 6-dichlorophenol-indophenol titrimetric method [17]. Three biological replicates were used for each sample, and the vitamin C content was expressed as mg/100g FW.

### 2.4. Total Anthocyanin, Total Phenol, and Flavonoid Determination

The contents of total anthocyanin, total phenol, and flavonoids were tested following the method previously detailed by Li et al. in [19]. In brief, 3 g of each fresh tissue was ground, and total anthocyanins were extracted with HCl/methanol (1:99, *v*/*v*) in the dark for 12 h. The supernatants of total anthocyanin, total phenol, and flavonoids were determined using a UV5100H UV-V spectrophotometer at 530 nm, 765 nm, and 510 nm. The results were expressed as the mean of three replicates (mg/kg FW).

### 2.5. Analysis of Anthocyanin Derivatives in Peaches Extracts by HPLC (High Performance Liquid Chromatography)

The standards of delphinidin-3-*O*-glucoside (Dp), cyanidin-3-*O*-glucoside (Cy), and pelargonidin-3-*O*-glucoside (Pg) (Solarbio Science & Technology Co., Ltd., Beijing, China) were used to detect the derivatives of anthocyanin using an LC-100 HPLC (Wufeng Scientific Instrument Co., Ltd., Shanghai, China). The peach fruit samples were extracted according to Lu et al. [20]. The column used was a Brisa LC2 C18 (4.6 mm × 250 mm, 5 μm) (Teknokroma Co., Barcelona, Spain) with a temperature of 45 °C, injection volume of 20 μL, and a detection wavelength of 520 nm. The gradient elution conditions were: (1) 0 min→15 min, 6%A→30%A; (2) 15 min→16 min 30 s, 30%A→50%A; (3) 16 min 30 s→18 min, 50%A→60%A; (4)18 min→20 min, 60%A→6%A.

### 2.6. Statistical Analysis

All values were shown as the mean ±SD in three biological replicates. The mean was computed using Microsoft Excel 2010 (Microsoft Co., Redmond, WA, USA). The STDEV function was used to calculate the standard deviation. Figures and line graphs were made using Graph Pad Prism 8 (Graph Pad Software Inc., San Diego, CA, USA). Statistical analysis was performed using the SPSS V13.0 software (SPSS Inc, Chicago, IL, USA). One-way ANOVA analysis was used to find significant statistical differences between treatments. Duncan’s test was used to compare the means at *p* < 0.05 (significant differences).

## 3. Results

### 3.1. Effect of Treatment on Good Fruit Rate, Fruit Firmness, and Flesh Browning Rate of Peach Fruit

Good fruit rate and fruit firmness showed a decreasing tendency during the storage period. No changes in good fruit rate in C treatment occurred during the cold storage (Figure 3a). After storage for 50 days, the rate in C treatment had the maximum value of 100%, which was 11.11%, 22.22% and 11.11% higher than that in A, B, and D treatments, respectively. It was also found that fruit firmness (Figure 3b) in the C treatment had significant differences (*p* < 0.05) compared to that in A, B, and D on the 10th, 40th and 50th days.

All treatments showed flesh browning after storage for 40 days (Figure 3). After 50 days in storage, the flesh browning rate in C treatment had a minimum value of 11.11%, which was 44.44%, 44.44%, and 33.33% lower than that in A, B, and D treatments, respectively.

### 3.2. Effect of Treatment on TSS, Soluble Sugar, Organic Acid, and Vitamin C

The TSS content showed an increasing trend at the beginning of postharvest and then declined (Figure 4). After storage for 50 days, the TSS content in the A, B, and C treatments were lower than that in the D treatment. The TSS content in the D treatment showed significant differences (*p* < 0.05) compared with that in the A, B, and C treatments on the 50th day. The average TSS content was 12.52% during the storage time.

The soluble sugar content of all tested parts first showed an increase and then a decrease with the increase in storage time (Figure 5a–c). The soluble sugar content of the peel and white flesh in the D treatment had the maximum values of 4.31% and 4.84%, respectively, and in the C treatment of red flesh, it was 3.83% at 50 days after storage. The significant differences (*p* < 0.05) in soluble sugar content of peel in the D treatment were observed and compared with that in A, B, and C treatments at the 10th, 30th, 40th and 50th days of storage. The red and white flesh in the D treatment showed significant differences in soluble sugar content (*p* < 0.05) compared with those in the A, B and D treatments on the 20th, 30th, 40th and 50th days of storage. The average soluble sugar content of peel, red flesh and white flesh were 3.34%, 3.58% and 3.68%, respectively.

The organic acid and vitamin C content in all treatments showed a downward trend during the storage period (Figure 5d–i). After 50 days of cold storage, the organic acid content in A, B, and D treatments (0.16%, 0.23%, and 0.25%, respectively), had lower contents in their peel than those of the C treatment (0.28%). The organic acid content of the C treatment in peel was significantly higher than in the A, B, and D treatments (*p* < 0.05) on the 30th, 40th and 50th days of storage. The organic acid content of the C treatment in red and white flesh was significantly higher than in the A, B, and D treatments (*p* < 0.05) on the 20th, 30th, 40th and 50th days of storage. The average soluble sugar content of peel, red flesh and white flesh were 3.34%, 3.58% and 3.68%, respectively. The vitamin C content of peel in the C treatment could not be detected after storage for 50 days, while for the other three treatments, the vitamin C content could not be detected after storage for 40 days. The vitamin C contents of both red and white flesh in the C treatment were higher than that in the other three treatments after storage for 50 days. The Vitamin C content of peel in the A treatment had significant differences (*p* < 0.05) compared to that in the B, C and D treatments on the 20th, 30th and 40th days of storage. The white flesh in the A treatment showed significant differences (*p* < 0.05) compared to that in the B, C, and D treatments on the 20th, 30th and 40th days of storage. The average vitamin C content of peel, red flesh and white flesh were 0.29%, 0.37% and 0.29%, respectively.

### 3.3. Effect of Treatment on Total Phenol and Flavonoid

As the storage time proceeded, the total phenol and flavonoid content showed an increasing tendency at the beginning of storage, and then decreased (Figure 6a–f). After storage for 50 days, fruits in the C treatment (1.78 mg/g FW) showed higher total phenol content in the peel compared to A, B, and D treatments (1.69 mg/g FW, 1.63 mg/g FW and 1.62 mg/g FW, respectively). The total phenol content of peel in the C treatment had significant differences (*p* < 0.05) compared to that in the A, B, and D treatments on the 20th, 40th and 50th days of storage. Red flesh in D treatment had significant differences (*p* < 0.05) compared to that in A, B and C treatments on the 20th, 30th, 40th and 50th days after storage. White flesh in A treatment had significant differences (*p* < 0.05) compared to that in B, C, and D treatments on the 10th, 20th, 30th and 50th days of storage. The average total phenol contents were 2.05 mg/g FW, 0.62 mg/g FW and 0.54 mg/g FW in peach peel, red flesh and white flesh, respectively, during the storage time. The C treatment (1.86 mg/g FW) showed a higher flavonoid content in the peel compared with A (1.51 mg/g FW), B (1.35 mg/g FW), and D treatments (1.38 mg/g FW). The flavonoid content of peel in the C treatment had significant differences (*p* < 0.05) compared to that in the A, B, and D treatments on the 10th, 20th, 40th and 50th days of storage. The flavonoid content of the red flesh and the white flesh showed a minor discrepancy among the four treatments; the average flavonoid contents were 2.02 mg/g FW, 0.22 mg/g FW and 0.22 mg/g FW in peach peel, red flesh and white flesh, respectively, during the storage time.

### 3.4. Effect of Treatment on Total Anthocyanin and Identities of Anthocyanin in the Peach Extract

In general, the total anthocyanin content first increased and then decreased over storage time (Figure 7a–c). The peel and white flesh in the C treatment showed a higher total anthocyanin content (19.37 mg/kg FW and 0.83 mg/kg FW, respectively), compared to other treatments after storage, and red flesh in the D treatment showed a maximum value of 13.01 mg/kg FW. The anthocyanin content of fruits in the C treatment had significant differences (*p* < 0.05) compared to that in the A, B, and D treatments on the 20th, 30th, 40th and 50th days of storage. The total average anthocyanin contents were 20.77 mg/kg FW, 20.66 mg/kg FW and 1.25 mg/kg FW in peach peel, red flesh and white flesh, respectively, during the storage time.

In the present study, two principal anthocyanins (cyanidin and delphinidin) were detected in the peach peel, and only one anthocyanin (cyanidin) was detected in the flesh extract by HPLC analysis at 520 nm (Figure 8).

In general, the content of cyanidin and delphinidin in peel and flesh decreased with storage time (Figure 7d–g). After the fruits were stored for 50 days, the peel and red flesh in the C treatment showed a higher cyanidin content (33.45 mg/kg FW and 40.42 mg/kg FW, respectively), compared with other treatments. The cyanidin content of peel in the C treatment had significant differences (*p* < 0.05) compared to that in the A, B, and D treatments on the 10th, 20th, 30th, 40th and 50th days of storage, respectively, and the cyanidin content of red flesh in C treatment had significant differences (*p* < 0.05) compared to that in A, B and C treatments at the 50th day of storage. After storage for 30 days, the cyanidin content of white flesh in the C treatment could not be detected, while in the other three treatments, the cyanidin content could not be detected after storage for 20 days. After the fruits were stored for 50 days, the C treatment (1.82 mg/kg FW) showed a higher delphinidin content compared to the A, B, and D treatments (1.48 mg/kg FW, 0.71 mg/kg FW and 1.41 mg/g FW, respectively). The delphinidin content of peel in the C treatment had significant differences (*p* < 0.05) compared to that in the A, B, and D treatments on the 10th, 20th, 40th and 50th days of storage, respectively. The average cyanidin content were 60.29 mg/kg FW, 50.05 mg/kg FW and 0.18 mg/kg FW in peach peel, red flesh and white flesh, respectively, during the storage time. The average pelargonin content was 2.09 mg/kg FW.

## 4. Discussion

### 4.1. Phytochemical Profiles of Different Peach Part

For years it has been proven that phytochemicals are not uniformly distributed in the fruit tissue, most of them being concentrated in the peel; the peel also contains higher amounts of phenolic compounds and other antioxidants than the flesh tissue [2]. In this experiment, we found that the anthocyanin, total phenol and flavonoids in peach peel were higher than those in the fleshy parts, and the total anthocyanin and cyanidin content of the red flesh near the stone of the peach was higher than that in the white flesh (far from the stone). This was also confirmed by Zhao et al. who found that anthocyanins were detected in higher amounts in the peach peel, and small amounts of pigments were also detected in the peach flesh, particularly in the tissues near the stone [21].

### 4.2. The Quality and Characteristics of ‘Yonglian No.1’ Peach

The results showed that with the increase in maturity, the fruit firmness of ‘Yonglian No.1’ peaches showed a decreasing tendency, the color of the peel began to deepen, and the vitamin C content showed a decreasing tendency, while the TSS and soluble sugar of peel, cyanidin, pelargonidin, total anthocyanin, and total phenol of flesh, and flavonoids showed an increasing tendency. Similar results were also reported by Zhou et al. [22]. The main anthocyanin in the peach peel was cyanidin, a small amount of pelargonidin was found in the peel, which was the main anthocyanin in the peach flesh. These results were supported by previous research [23,24,25].

Peach cultivars can be classified according to their phenolic profile, and this profile also determines how beneficial a certain cultivar is for human health [2]. The average total phenol concentrations reach values of up to 0.16 mg/g FW in 257 peach germplasms [26], while the concentration values of ‘Yonglian No.1’ were 2.05 mg/g FW and 0.58 mg/g FW in peach peel and flesh, respectively. It was also found that the average flavonoid content of five different peach varieties was 0.13 mg/g FW [27], while it was 2.02 mg/g FW and 0.22 mg/g FW in ‘Yonglian No.1’ peach peel and flesh, respectively. Average cyanidin content in the ‘Yonglian No.1’ peach, 60.29 mg/kg FW in the peel and 25.12 mg/kg FW in flesh, was much higher than that of white-flesh peach cultivars (39.90 mg/kg FW and 21.50 mg/kg FW, respectively) [5].

The storage characteristics of ‘Yonglian No.1’ peaches are better than for the ‘Yingzui honey peach’ [28] and ‘Xinfengmilu’ [29], as shown in Table 1, with a higher percentage of good fruit rate, fruit firmness, TSS, and longer storage time during cold storage.

### 4.3. Effect of MAP on Peach Fruit Quality during Cold Storage

Previous studies have investigated the effect of MAP on phytochemicals in fruits. In citrus fruits, the use of MAP has reduced the incidence of peel disorders and chilling injury [14]. In plums, MAP maintained high-quality attributes and delayed ripening and development of bioactive compounds during storage [30,31]. In strawberries and blueberries, MAP delays the softening speed and prolongs the storage period [32,33]. MAP can effectively maintain the sensory quality and improve the commercialization of grapes [34]. As a typical climacteric fruit, the preservation period of the ‘Yonglian No.1’ peach is short at room temperature. In peaches and nectarines, MAP slows down the respiration rate of fruits and retards the decrease in titratable acidity values, maintaining the fruit sugar and soluble solids content, flesh firmness, vitamin C and juice content, and slows deterioration through decreasing fruit injury and browning development [15]. This study demonstrated that MAP is an excellent addition to 4 ± 1 °C temperature management for extending the storage life of ‘Yonglian No.1’ peaches produced in Fujian province. The storage quality maintained by MAP + washing extended the storage life, minimized the loss of fruit firmness and color, and provided fruit which had a higher content of soluble sugar, organic acid, vitamin C, and phenol at more than 40 days of storage. Fruit held in MAP for 40 days or longer were of a higher eating quality after ripening than AP.

## 5. Conclusions

The ‘Yonglian No.1’ peach is a very nutritious fruit, endowed with significant biochemicals. It can be used as an excellent fresh peach variety, which is of great significance in regulating the supply of off-season fruits, meeting the needs of people’s daily lives. Storage by MAP is a valuable approach to maintain the health-promoting compounds of peach fruits.

## Figures and Tables

**Figure 1 molecules-27-06319-f001:**
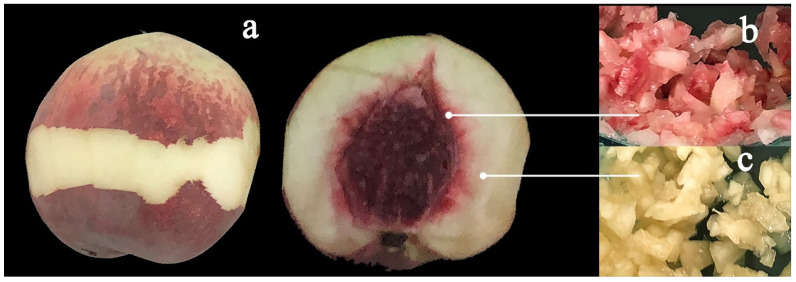
The peel taken from the equator of peach fruit (**a**). Red flesh (**b**) and white flesh (**c**) from the peach fruit.

**Figure 2 molecules-27-06319-f002:**
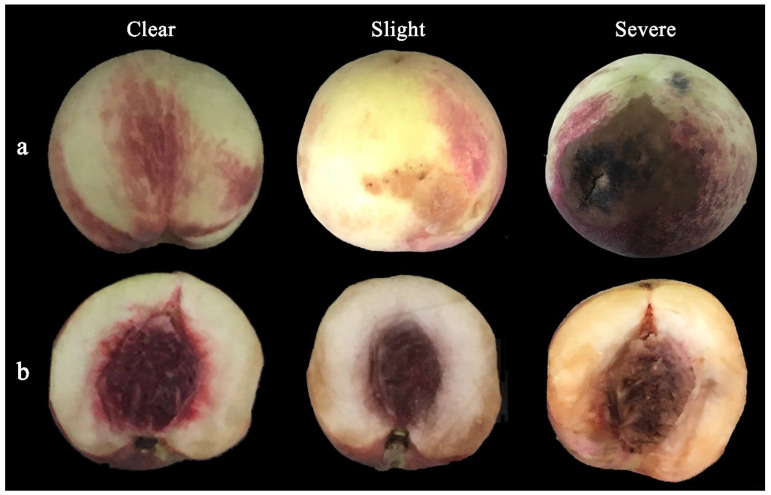
Appearance (**a**) and cut longitudinally (**b**) to assess for good fruit, flesh browning and rotten fruits (From left to right).

**Figure 3 molecules-27-06319-f003:**
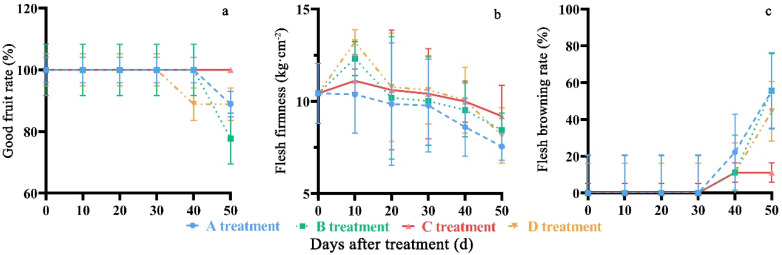
Changes in good fruit rate (**a**), fruit firmness (**b**) and flesh browning rate (**c**) of ‘Yonglian No.1’ peach fruits during storage in four treatments. Vertical bars represent the standard errors. The different lower case letters at each time point indicate a significant difference at *p* < 0.05 using Duncan’s multiple range tests.

**Figure 4 molecules-27-06319-f004:**
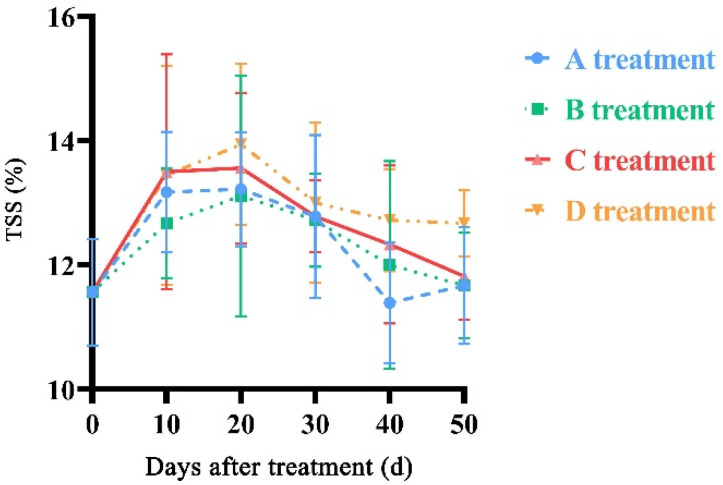
Changes in TSS of ‘Yonglian No.1’ peach fruits during storage in four treatments. Vertical bars represent the standard errors. The different lower case letters at each time point indicate significant difference at *p* < 0.05 using Duncan’s multiple range tests.

**Figure 5 molecules-27-06319-f005:**
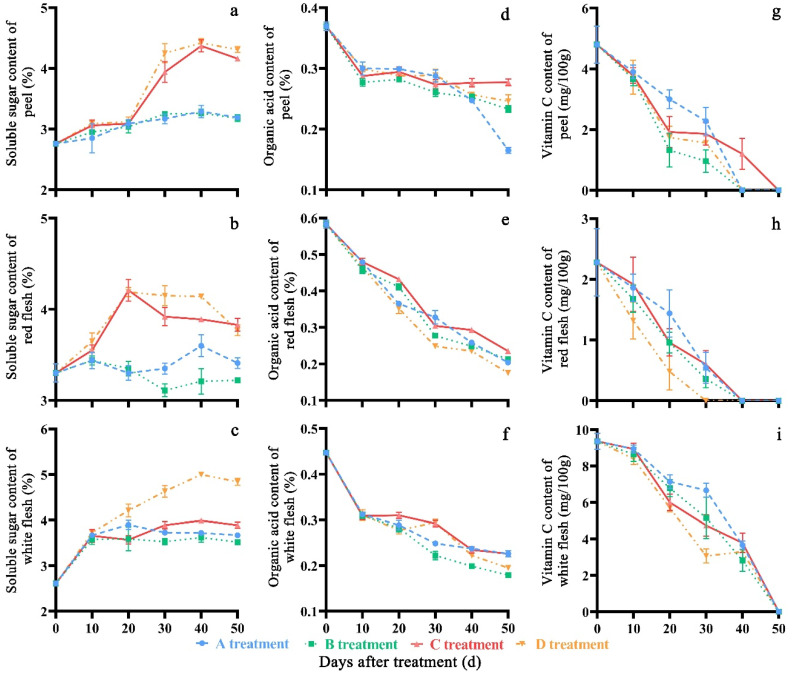
Changes in soluble sugar (**a**–**c**), organic acid (**d**–**f**) and vitamin C (**g**–**i**) content of ‘Yonglian No.1’ peach fruits during storage in four treatments. Vertical bars represent the standard errors. The different lower case letters at each time point indicate significant difference at *p* < 0.05 using Duncan’s multiple range tests.

**Figure 6 molecules-27-06319-f006:**
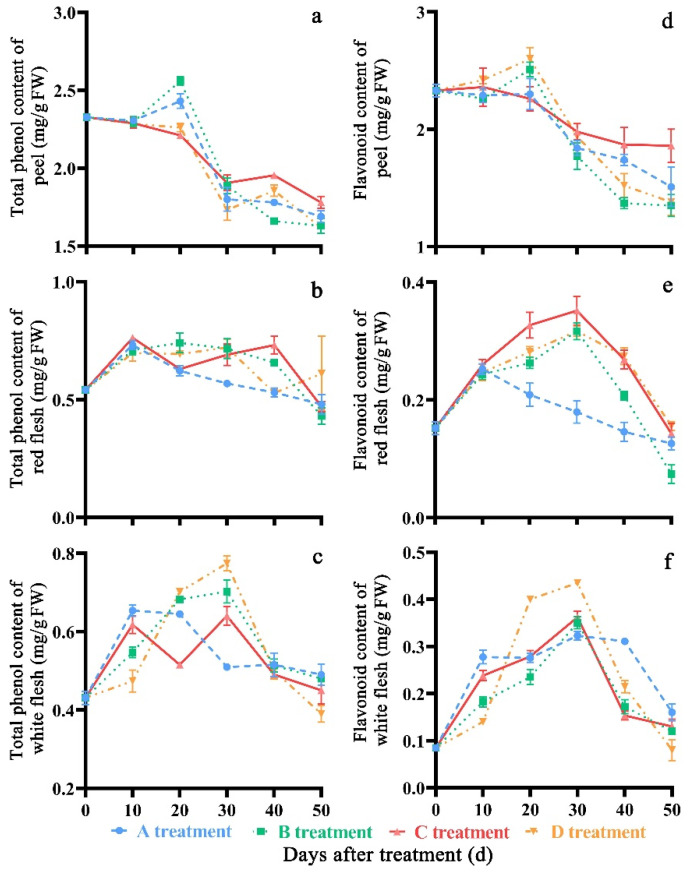
Changes in total phenol (**a**–**c**) and flavonoid (**d**–**f**) content of ‘Yonglian No.1’ peach fruits during storage in four treatments. Vertical bars represent the standard errors. The different lower case letters at each time point indicate significant difference at *p* < 0.05 using Duncan’s multiple range tests.

**Figure 7 molecules-27-06319-f007:**
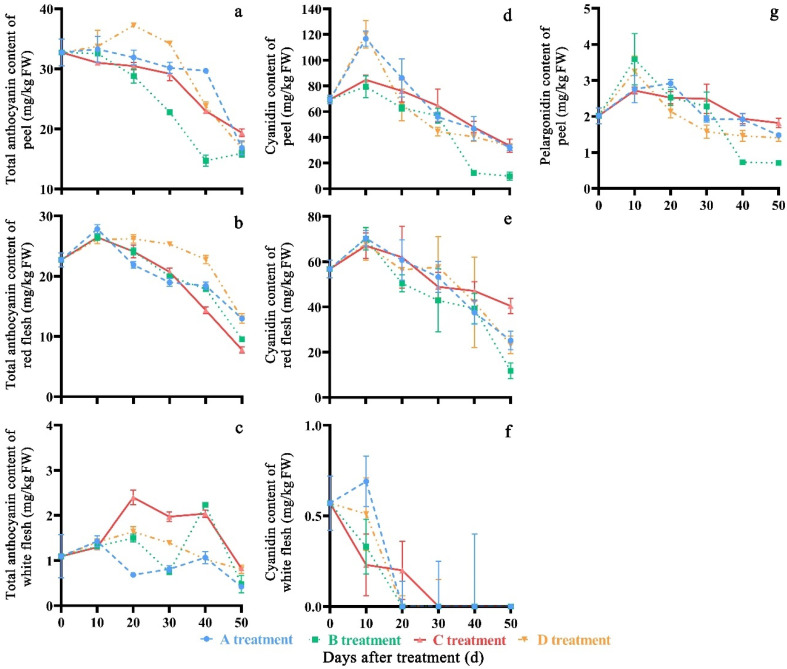
Changes in total anthocyanin (**a**–**c**), cyanidin (**d**–**f**) and pelargonidin (**g**) content of ‘Yonglian No.1’ peach fruits during storage in four treatments. Vertical bars represent the standard errors. The different lower case letters at each time point indicate significant difference at *p* < 0.05 using Duncan’s multiple range tests.

**Figure 8 molecules-27-06319-f008:**
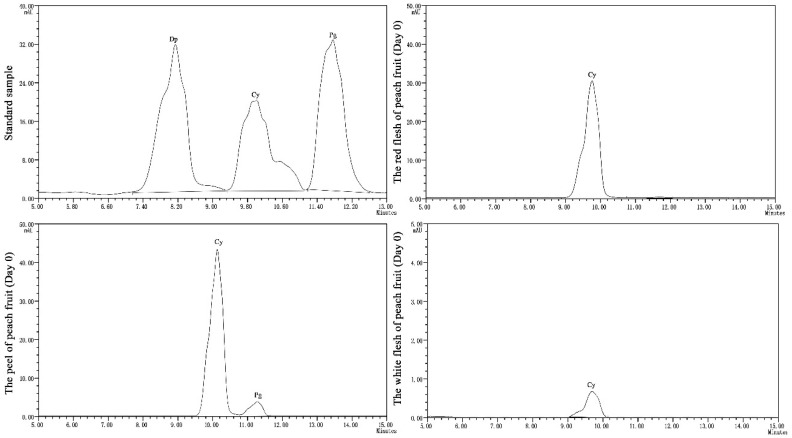
High performance liquid chromatograms (HPLC) of anthocyanin from ‘Yonglian No.1’ peach fruits (Dp: delphinidin-3-*O*-glucoside, Cy: cyanidin-3-*O*-glucoside, Pg: pelargonidin-3-*O*-glucosided).

**Table 1 molecules-27-06319-t001:** Storage quality index of different peach varieties.

	Good Fruit Rate (%)	Fruit Firmness (kg/cm^2^)	TSS (%)	Storage Time(d)	Storage Temperature (°C)	References
‘Yonglian No.1’	100	9.19	12.67	50	4 ± 1	-
‘Yingzui honey peach’	84.5	8.2	9.51	40	2 ± 1	[26]
‘Xinfengmilu’	58.3	9.00	-	35	3 ± 1	[27]

## Data Availability

The data presented in this study are available on request from the corresponding author.

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
