# Peer review of "Changes in Fruit Quality Phytochemicals of Late-Mature Peach ‘Yonglian No.1’ during Storage"

_molecules, 2022, doi:10.3390/molecules27196319_

Round 1
Reviewer 1 Report
The authors try to present a paper that stands out for its title, but reading the article reveals too many gaps.
The introduction talks about the fruit in question, but little is mentioned about how MAP can affect fruits in general.
In Materials and Methods, the characteristics of the MAP are not mentioned, nor are the characteristics of the containers used. For the type of analysis, it is relevant to place it.
As for the results, these are fine, but the discussion practically does not exist. The comments have more to do with going up or down, but they do not indicate what the values could be due to. There is a lack of analysis and that it is in accordance with the atmospheres to which the fruits were subjected. Therefore, it is important to include the characteristics of the MAP.
The Conclusions do not indicate anything new.
The paper, as it stands, should not be published.
Author Response
Point 1: The authors try to present a paper that stands out for its title, but reading the article reveals too many gaps.
The introduction talks about the fruit in question, but little is mentioned about how MAP can affect fruits in general.
Response 1: It was supplemented in the ‘Introduction’ part.
Modified atmosphere packaging (MAP) was used to replenish low-temperature regulation by lowering O2 levels and increasing CO2 and humidity levels around the fruit or other produce, to delay ripening and suppress decay in fresh fruit, inhibit the degradation of chlorophyll and other pigments, and maintain the organoleptic properties and fresh-like characteristics of the fruit
Point 2: In Materials and Methods, the characteristics of the MAP are not mentioned, nor are the characteristics of the containers used. For the type of analysis, it is relevant to place it.
Response 1: It have been mentioned in the ‘Merials and Methods’ (page2) as follows:
AP, nine pores were punched uniformly on the surface of the PE bag (280mm × 400mm, 9.40g, 0.03 mm thick) with a punch (8mm diameter of each); MAP, no pores were punched on the surface of PE bag. And packaging batches for each treatment were as follows:
A treatment (MAP): the peach fruit without washing and the PE bag without pore;
B treatment (AP): the peach fruit without washing and the PE bag with pores;
C treatment (MAP + Washing): the peach fruit washing with tap water and the PE bag without pore;
D treatment (AP + Washing): the peach fruit washing with tap water and the PE bag with pores.
Point 3: As for the results, these are fine, but the discussion practically does not exist. The comments have more to do with going up or down, but they do not indicate what the values could be due to. There is a lack of analysis and that it is in accordance with the atmospheres to which the fruits were subjected. Therefore, it is important to include the characteristics of the MAP.
The Conclusions do not indicate anything new.
The paper, as it stands, should not be published.
Response 3: The results and conclusion were supplemented.
4.3 Effect of MAP on peach fruit quality during cold storage
Previous studies have investigated the effect of MAP on phytochemicals in fruits. In citrus fruits, use of MAP has reduced the incidence of peel disorders and chilling injury [14]. In plums, MAP maintained high quality attributes and delayed ripening and development of bioactive compounds during storage [29, 30]. In strawberries and blueberries, MAP delayed the softening speed and prolonged the storage period [31, 32]. MAP can effectively maintain the sensory quality and improve the commercialization of grapes [33]. As a typical climacteric fruit, the preservation period of ‘Yonglian No.1’ peach was short at room temperature. In peaches and nectarines, MAP slowed down the respiration rate of fruits and retarded the decrease in titratable acidity values, maintained the fruit sugar and soluble solids content, flesh firmness, vitamin C and juice content, and slowed deterioration through decreasing fruit injury and browning development [15]. This study demonstrated that MAP is an excellent addition to 4±1 ℃ temperature management for extending the storage life of ‘Yonglian No.1’ peach produced in Fujian province. The storage quality maintained by MAP + washing extended the storage life, minimized the loss of fruit firmness and color, and provided fruit that were higher content of soluble sugar, organic acid, vitamin C, phenol to more than 40 days. Fruit held in MAP for either more than 40 days had higher eating quality after ripening than AP.
- Conclusion
‘Yonglian No.1’ peach is a very nutritious fruit and endowed with significant biochemicals. It can be used as an excellent fresh peach variety, which is of great significance to regulate the supply of off-season fruits, meet the needs of people's daily life. Storage by MAP is a valuable approach to maintain the health-promoting compounds of peach fruits.
Reviewer 2 Report
The manuscript is interesting, and written very neat. It can be considered after minor revision. Some hints for authors.
1. Please write capital letters only in starting sentence sodium hydroxide small like on page 4.
2. Please compare with other fruit for example pepper and grape
3. Please add the Anova formula to improve the clarity of the calculation
4. Try to extend conclusions
Author Response
The manuscript is interesting, and written very neat. It can be considered after minor revision. Some hints for authors.
Point 1: Please write capital letters only in starting sentence sodium hydroxide small like on page 4.
Response 1: Thank you for pointing out our mistakes, we have modified them with “Track Changes” function.
Point 2: Please compare with other fruit for example pepper and grape
Response 2: It was supplemented in the ‘4.3. Effect of MAP on peach fruit quality during cold storage’.
4.3 Effect of MAP on peach fruit quality during cold storage
Previous studies have investigated the effect of MAP on phytochemicals in fruits. In citrus fruits, use of MAP has reduced the incidence of peel disorders and chilling injury [14]. In plums, MAP maintained high quality attributes and delayed ripening and development of bioactive compounds during storage [30, 31]. In strawberries and blueberries, MAP delayed the softening speed and prolonged the storage period [32, 33]. MAP can effectively maintain the sensory quality and improve the commercialization of grapes [34]. As a typical climacteric fruit, the preservation period of ‘Yonglian No.1’ peach was short at room temperature. In peaches and nectarines, MAP slowed down the respiration rate of fruits and retarded the decrease in titratable acidity values, maintained the fruit sugar and soluble solids content, flesh firmness, vitamin C and juice content, and slowed deterioration through decreasing fruit injury and browning development [15]. This study demonstrated that MAP is an excellent addition to 4±1 ℃ temperature management for extending the storage life of ‘Yonglian No.1’ peach produced in Fujian province. The storage quality maintained by MAP + washing extended the storage life, minimized the loss of fruit firmness and color, and provided fruit that were higher content of soluble sugar, organic acid, vitamin C, phenol to more than 40 days. Fruit held in MAP for either more than 40 days had higher eating quality after ripening than AP.
Point 3: Please add the Anova formula to improve the clarity of the calculation
Response 3: It was supplemented in the ‘2.6. Statistical analysis’. STDEV function was used to calculate the standard deviation.
Point 4: Try to extend conclusions
Response 4: It was supplemented.
‘Yonglian No.1’ peach is a very nutritious fruit and endowed with significant biochemicals. It can be used as an excellent fresh peach variety, which is of great significance to regulate the supply of off-season fruits, meet the needs of people's daily life. Storage by MAP is a valuable approach to maintain the health‐promoting compounds of peach fruits.
Reviewer 3 Report
The article is interesting, the changes in yonlglian peaches stored in MAP under refrigeration for 50 days are presented.
The most significant changes for the different parameters are presented in the results.
The methodology is described briefly and adequately.
Regarding the results, the discussions in the article could be improved, for some figures there is a good discussion and treatment of results, but in others they are barely mentioned.
Finally, the conclusions are very brief and I think they could be more explicit to improve the reader's understanding.

Author Response
The article is interesting, the changes in yonlglian peaches stored in MAP under refrigeration for 50 days are presented.
The most significant changes for the different parameters are presented in the results.
The methodology is described briefly and adequately.
Point 1: Regarding the results, the discussions in the article could be improved, for some figures there is a good discussion and treatment of results, but in others they are barely mentioned.
Response 1: It was supplemented in the ‘Results’.
And the average TSS content was 12.52% during the storage time.
The average soluble sugar content of peel, red flesh and white flesh were 3.34%, 3.58% and 3.68%, respectively.
The average soluble sugar content of peel, red flesh and white flesh were 3.34%, 3.58% and 3.68%, respectively.
The average vitamin C content of peel, red flesh and white flesh were 0.29%, 0.37% and 0.29%, respectively.
And the average total phenol content were 2.05 mg/g FW, 0.62 mg/g FW and 0.54 mg/g FW in peach peel, red flesh and white flesh during the storage time.
And the average flavonoid content were 2.02 mg/g FW, 0.22 mg/g FW and 0.22 mg/g FW in peach peel, red flesh and white flesh during the storage time.
And the average total anthocyanin content were 20.77 mg/kg FW, 20.66 mg/kg FW and 1.25 mg/kg FW in peach peel, red flesh and white flesh during the storage time.
And the average cyanidin content were 60.29 mg/kg FW, 50.05 mg/kg FW and 0.18 mg/kg FW in peach peel, red flesh and white flesh during the storage time. The average pelargonin content were 2.09 mg/kg FW.
Point 2: Finally, the conclusions are very brief and I think they could be more explicit to improve the reader's understanding.
Response 2: The conclusion and results were supplemented.
4.3 Effect of MAP on peach fruit quality during cold storage
Previous studies have investigated the effect of MAP on phytochemicals in fruits. In citrus fruits, use of MAP has reduced the incidence of peel disorders and chilling in-jury [14]. In plums, MAP maintained high quality attributes and delayed ripening and development of bioactive compounds during storage [30, 31]. In strawberries and blueberries, MAP delayed the softening speed and prolonged the storage period [32, 33]. MAP can effectively maintain the sensory quality and improve the commerciali-zation of grapes [34]. As a typical climacteric fruit, the preservation period of ‘Yongli-an No.1’ peach was short at room temperature. In peaches and nectarines, MAP slowed down the respiration rate of fruits and retarded the decrease in titratable acidi-ty values, maintained the fruit sugar and soluble solids content, flesh firmness, vitamin C and juice content, and slowed deterioration through decreasing fruit injury and browning development [15]. This study demonstrated that MAP is an excellent addi-tion to 4±1 ℃ temperature management for extending the storage life of ‘Yonglian No.1’ peach produced in Fujian province. The storage quality maintained by MAP + washing extended the storage life, minimized the loss of fruit firmness and color, and provided fruit that were higher content of soluble sugar, organic acid, vitamin C, phe-nol to more than 40 days. Fruit held in MAP for either more than 40 days had higher eating quality after ripening than AP.
- Conclusion
‘Yonglian No.1’ peach is a very nutritious fruit and endowed with significant biochemicals. It can be used as an excellent fresh peach variety, which is of great significance to regulate the supply of off-season fruits, meet the needs of people's daily life. Storage by MAP is a valuable approach to maintain the health-promoting compounds of peach fruits.